# Three-Dimensional Model of Dorsal Root Ganglion Explant as a Method of Studying Neurotrophic Factors in Regenerative Medicine

**DOI:** 10.3390/biomedicines8030049

**Published:** 2020-03-03

**Authors:** Polina Klimovich, Kseniya Rubina, Veronika Sysoeva, Ekaterina Semina

**Affiliations:** 1Laboratory of molecular endocrinology, National Cardiology Research Center Ministry of Health of the Russian Federation, Institute of Experimental Cardiology, Moscow 121552, Russia; lex2050@mail.ru; 2Faculty of Medicine, Lomonosov Moscow State University, Moscow 119991, Russia; veronikasysoeva@gmail.com; 3Laboratory of Morphogenesis and Tissue Reparation, Faculty of Medicine, Lomonosov Moscow State University, Moscow 119991, Russia; rkseniya@mail.ru

**Keywords:** axon regeneration, neurotrophic factors, Nestin-GFP mice, neurite outgrowth, 3D dorsal root ganglia explants, NGF, BDNF, GDNF, Matrigel

## Abstract

Neurotrophic factors play a key role in the development, differentiation, and survival of neurons and nerve regeneration. In the present study, we evaluated the effect of certain neurotrophic factors (NGF, BDNF, and GDNF) on axon growth and migration of Nestin-green fluorescent protein (GFP)-positive cells using a 3D model of dorsal root ganglion (DRG) explant culture in Matrigel. Our method generally represents a convenient model for assessing the effects of soluble factors and therapeutic agents on axon growth and nerve regeneration in R&D studies. By analyzing the DRG explants in ex vivo culture for 21 days, one can evaluate the parameters of neurite outgrowth and the rate of cell migration from the DRG explants into the Matrigel. For the current study, we used Nestin-GFP-expressing mice in which neural precursors express Nestin and the green fluorescent protein (GFP) under the same promoter. We revealed that GDNF significantly (two fold) stimulated axon outgrowth (*p* < 0.05), but not BDNF or NGF. It is well-known that axon growth can be stimulated by activated glial cells that fulfill a trophic function for regenerating nerves. For this reason, we evaluated the number of Nestin-GFP-positive cells that migrated from the DRG into the Matrigel in our 3D ex vivo explant model. We found that NGF and GDNF, but not BDNF, stimulated the migration of Nestin-GFP cells compared to the control (*p* < 0.05). On the basis of the aforementioned finding, we concluded that GDNF had the greatest stimulating potential for axon regeneration, as it stimulated not only the axon outgrowth, but also glial cell migration. Although NGF significantly stimulated glial cell migration, its effect on axon growth was insufficient for axon regeneration.

## 1. Introduction

Peripheral nerve injuries remain one of the most significant causes of disability and have a marked impact on patients’ everyday lives [1]. Despite meticulous surgical techniques and non-surgical treatments, fully functional recovery is rarely achieved. Therefore, there is a demand for new approaches to stimulate effective peripheral nerve regeneration after injury [2,3,4,5].

Regeneration in the nervous system is difficult to investigate due to the high complexity of neuronal network and the difficulty in culturing neural cells. 2D-culturing systems remain the predominant approach used for screening and testing new soluble factors and therapeutic agents due to their feasibility [6,7]. Among the disadvantages of 2D cultures is the lack of 3D extracellular matrix (ECM) that surrounds the cells and creates the microenvironment in vivo [6,8]. The lack of the proper cell–cell and cell–matrix contacts results in change of gene and protein expression profiles [9]. These can adversely affect cell morphology, cell cycle progression, cell survival, and intracellular signaling, potentially resulting in false positive or false negative data acquisition [8,10].

In 2D culturing systems, the bioavailability of substances is difficult to evaluate [11]. All cells in a monolayer in 2D culturing conditions simultaneously receive the same dose of a tested agent, whereas in vivo there is a gradient of a tested agent that penetrates the matrix and several layers of cells in a tissue [12]. 3D cultures allow for the assessment of the parameters of a gradual diffusion of tested agents through the gel [6,11].

Primary neural cell cultures represent a modern challenge in cell biology, as mature neurons do not undergo cell division in vitro, making it complicated to obtain a sufficient number of cells for experiments [13]. Cell survival is usually low, and neural cell cultures from DRG can be maintained for only 5–7 days, provided that soluble factors that increase survival are added [14]. This is one of the complications in result interpretation when analyzing the effects of various agents on cell migration and neurite formation. Explants, isolated and cultured ex vivo in 3D conditions, present a convenient model because they retain tissue-specific characteristics and long-term interactions between various cell types (neurons and glial cells), ECM, and biologically active molecules. This method can be used not only in the neurological field but also for cardiology, cancer biology, and other fields [15].

Neurotrophic factors are well-known regulators of cell proliferation, survival, migration, and differentiation in the nervous system [16]. They are constitutively expressed both in the embryonic peripheral and central nervous systems [17]. The expression of neurotrophic factors in the adult organism is sustained at a low level and increases upon injury, suggesting an important role in nerve regeneration [18]. Axon regeneration in the peripheral nervous system (PNS) is largely provided by Schwann cells, which actively proliferate, secrete growth factors, and produce large amounts of neurotrophic factors upon injury [19,20].

Despite the accumulated data on the important role of neurotrophic factors in PNS regeneration, the problem of full nerve recovery and clinical outcome is far from being solved. The therapy using neurotrophic factors is based on their capacity to promote survival of degenerating neurons; however, such therapy remains insufficient [21]. The development of new models for studying axonal regeneration, including neurotrophin-based therapy, can establish additional mechanisms of axonal growth and glial cell migration.

In the present study, we describe in detail a method of 3D culturing of dorsal root ganglia (DRG) explants in Matrigel ex vivo. This method was developed by us previously [15], and has demonstrated high reproducibility. We consider a 3D explant DRG culture ex vivo to be a relevant model resembling in vivo regeneration because it allows for the evaluation of the outgrowth of neurites and the migration of glial cells, which can be subjected to qualitative and quantitative analysis. Here, we tested the effects of neurotrophic factors (nerve growth factor (NGF), brain-derived neurotrophic factor (BDNF), and Glial cell-derived neurotrophic factor (GDNF)) on axon regeneration using Nestin-green fluorescent protein (GFP) mice. Besides the parameters of neurite outgrowth, this allowed us to describe migration of cells from the DRG. The designed protocol sustains the integrity of the isolated tissues and ensures the contact of neurons with glial cells and ECM, thus supporting neuron survival and physiological function. The current protocol includes the isolation, culturing, and immunofluorescent staining of DRG in Matrigel using specific antibodies. The protocol allows for the evaluation of the parameters of radial axonal growth, neurite branching, and glial cell migration in real time for 21 days. Additional administration of the tested agents into the Matrigel allows for the assessment of their effect on cell migration, axon growth, and branching. The procedure of 3D immunofluorescent staining of the whole mount DRG in Matrigel takes 7 days. The time required for visualization using confocal microscopy depends on the number of experimental points, and usually takes up to 3 h for high-resolution images.

## 2. Experimental Section

### 2.1. Equipment

Small animal operating table; gauze swab cloth, sterile (any supplier); stereomicroscope Olympus SZX16; Tungsten Carbide Iris Scissors straight, 4.5” (Braintree Scientific, Inc., Cat# SC-T 405); Dumont Inox Forceps, 45 degree, Inox, 109mm L (Braintree Scientific, Inc., Cat# FC-5005); Dumont Forceps, Inox, curved, 115mm L (Braintree Scientific, Inc., Cat# FC-5047); Micro Scissors, straight, 3.25” superfine blades (Braintree Scientific, Inc., Cat# SC-MS 152); SuperCut Surgical Scissors, straight, 5.5” sharp/blunt (Braintree Scientific, Inc., Cat# SCT-S 508); confocal laser scanning microscope Leica TCS SP5 (Leica microsystems, Wetzlar, Germany) or analogous.

### 2.2. Animals and Ethics Statement

The study was completed in accordance with the European convention for the Protection of Vertebrate Animals used for Experimental and other Scientific purposes (ETS 123). All routine procedures, including animal housing and care, were conducted in accordance with the revised Appendix A to ETS 123. All manipulations with animals were carried out in accordance with the requirements of Order No. 267 of the Ministry of Health of the Russian Federation “On 156 Approval of Laboratory Practice Rules” (no. 4809, 19 June 2003) and were approved by the local ethical committee in accordance with internal requirements of the Commission on Bioethics of the Faculty of Medicine of Lomonosov Moscow State University. We used 8–9 week Nestin-GFP male mice that were kindly provided to us by Grigori Enikolopov [22]. Nestin-GFP mice were used to isolate dorsal root ganglia (DRG). Before the experimental manipulation, animals were anesthetized by intraperitoneal injection with 400 μL of a 2.5% solution of 2,2,2-tribromomethanol (Sigmaaldrich, St. Louis, Missouri, USA).

### 2.3. Isolation and Culturing of DRG

Before all manipulations, the surgical instruments and cotton swabs were sterilized by autoclaving. The operating table was prepared by wiping the surface with ethanol 70%. A mouse was lethally anesthetized, decapitated, and the blood was carefully drained, paying special attention to protect the spine and avoiding fractures or ruptures of the spinal column. All manipulations on DRG isolation were carried out quickly, as the increase in time until the removal of DRG and placement of them into the culture media can reduce DRG viability.

A mouse was placed onto the operating table so that its limbs were maximally distant from the body, and the operating area was wiped and the skin from the back was removed. The skin was gently cut, avoiding the ingress of hair into the surgical field. Scissors and tweezers were used to release the spinal column from the muscle and connective tissues, avoiding any damaging of the nerves that emerge from the spinal column. The spinal canal was opened with the scissors and the vertebral arch along with the connecting muscles was removed; then, the spine was opened, freeing the spinal cord (Figure 1).

DRGs were located in the intervertebral foramina. DRGs were isolated using sterile surgical instruments for small laboratory animals under a stereomicroscope Olympus SZX16. The isolated tissues were immediately placed in a Petri dish containing sterile pre-chilled non-phenol red sterile Hank’s Balanced Salt Solution (HBSS) (Gibco, Cat# 14025–092) (Figure 2) and transferred into the laminar flow hood. All further manipulations were performed in sterile conditions. Drying of the DRGs should be avoided, as this can increase the stickiness of tissue to the forceps, making it difficult to attach them to the bottom of a coverglass. Four hours prior to the experiment, the frozen Matrigel growth factors reduced (Matrigel^TM^ Basement Membrane Matrix, BD, Cat# 354230, USA) was heated up to +4 °C on ice. For 3D tissue explant culture, DRGs were washed in sterile HBSS, placed in an 8-well sterile chambered borosilicate coverglass (Lab-Tek, Cat# 155411) and covered immediately with a drop (60 µL) of Matrigel and allowed to polymerize (Figure 3). An increase in the volume of Matrigel drop exceeding 70 µL and Matrigel bubble formation should be avoided. For 2D DRG cultures, the plates were covered with cold Matrigel and allowed to polymerize. DRGs were placed on polymerized Matrigel surface. After Matrigel polymerization, the wells with 2D and 3D DRG cultures were filled with a warm Roswell Park Memorial Institute medium (RPMI) 1640 culturing media containing GlutaMAX^T^ Supplement, Phenol Red (Gibco, Cat# 61870–010), antibiotic-antimycotic solution (100X, Gibco, Cat# 15240–096). Mouse NGF (100 ng/mL; Sigma-Aldrich, Cat# SRP4304), mouse BDNF (50 ng/mL; Abcam, Cat# ab9794), and mouse GDNF (50 ng/mL; Abcam, Cat# ab56286) or 10% bovine serum albumin (BSA), lyophilized powder, crystallized, ≥ 98.0% (GE; Sigma-Aldrich, Cat# 05470–5G) as a control were added to the culture media. DRG explants were cultured in 5% CO_2_ incubator at 37 °C for 14 days. DRGs were analyzed using whole mount immunofluorescent staining with antibodies (whole 3D mount assay) combined with confocal microscopy.

### 2.4. 3D Immunofluorescent Staining of DRG

The staining should be carried out at 37 °C to prevent Matrigel depolymerization. For 3D immunofluorescent staining, the wells with DRG were washed with warm HBSS (Gibco). The samples were fixed with 4% paraformaldehyde on HBSS for 24 h (Panreac). After fixation, the samples were washed in HBSS for 24 h and permeabilized with 1% Triton X-100 (TritonX-100, Peroxide Free, Panreac) for 24 h, and washed again in HBSS buffer for 24 h. Further, the samples were treated with 10% of normal donkey serum (Sigma-Aldrich), containing 1% Triton X-100, for 24 h to block non-specific antibody binding; washed in HBSS containing 1% Triton X-100 for 24 h; and incubated in the solution of the first antibodies rabbit anti-NF200 recognizing axon neurofilaments 200 kD (Abcam, Cat# ab8135) for 24 h. Next, the samples were washed in HBSS for 24 h, and incubated in the solution of the anti-rabbit second antibodies conjugated with fluorochrome AlexaFluor594 (1:500, Molecular Probes). Finally, the samples were placed in HBSS containing DAPI (Sigmaaldrich, 1:10,000) to visualize nuclei and incubated for 24 h. To prevent contamination, the samples were stored in HBSS solution containing 0.001% Sodium azide NaN_3_ (EMD Millipore, Cat# MSX0299–1). The number of outgrowing axons and of migrating Nestin-GFP-positive cells was evaluated using confocal microscopy with subsequent analysis with an automated image analysis program (ImageJ, National Institute of Health, Bethesda, MD, USA).

### 2.5. Isolation and Cultivation of the Primary Cell Culture from DRG

To obtain the primary cell culture, DRG from Nestin-GFP mice were isolated in sterile conditions, as described above, and placed in a 1 mg/mL collagenase solution (collagenase type I, 285 U/mg, Cat# 17100–017, Gibco, Waltham, MA, USA) prepared on Hank’s buffer. DRGs were incubated for 90 min at 37 °C. After that, samples were homogenized with an insulin syringe and centrifuged. The collagenase solution was removed and 1 mL of complete Dulbecco’s Modified Eagle Medium (DMEM) medium was added. Samples were pipetted to disaggregate the cells and applied to coverglass. Cells were left overnight in an incubator until complete adhesion and cultured in DMEM medium for at least 3 days prior to immunofluorescent staining.

### 2.6. Immunofluorescent Staining of Cells

Cells were washed with phosphate-buffered saline (PBS), fixed with 4% paraformaldehyde solution for 10 min, and washed in PBS. To block nonspecific binding, cells were treated with a 10% solution of normal donkey serum (Sigma Aldrich, USA) prepared on 5% bovine serum albumin (BSA) (Sigmaaldrich, St. Louis, MI, USA) in PBS for 1 h at room temperature. After washing, samples were incubated with primary goat anti-mouse antibodies against Glial fibrillary acidic protein (GFAP) (Abcam, Cat# ab53554) at 4 °C overnight, washed three times in PBS, and incubated with anti-goat second antibodies conjugated with fluorochrome AlexaFluor594 for 1 h. To visualize the nuclei, the samples were stained with DAPI (Sigma Aldrich, 1:10,000). After washing in PBS, the samples were enclosed in the nonfluorescent aqueous medium Aqua Poly Mount (Polysciences, Germany).

### 2.7. Microscopy

All manipulations with DRGs should be carried out at 37 °C to prevent Matrigel depolymerization. Images of unstained DRG were acquired using light microscopy (Zeiss Axiovert 25,Carl Zeiss Microscopy GmbH, Jena, Germany), whereas images of prestained DRG and of the primary cell cultures were acquired using confocal laser scanning microscope Leica TCS SP5 (Leica microsystems, Wetzlar, Germany) equipped with a PlanApo ×10, oil objective for DRG imaging and PlanApo ×63, 1.40 NA oil objective for cell imaging. DAPI, AlexaFluor594, and GFP fluorescence were sequentially excited using lasers with 405, 594, and 488 wavelengths, respectively. All images were captured with the same confocal gain and offset settings, and were further analyzed using ImageJ software. Images were compiled in Photoshop (version CS5, Adobe) to generate figures. Images are presented as an overlap of green, red, and blue staining. Video files were captured using LAS AF application software (Appendix A).

### 2.8. Statistical Analysis

Data were analyzed using SigmaPlot11.0 Software (Systat Software Inc. (SSI), San Jose, CA, USA). All normally distributed data with uniform group variances were analyzed using ANOVA with post-hoc Tukey’s honest significant difference test. Normally distributed data are presented as mean ± standard deviation. A statistically significant difference was considered for *p* < 0.05; *n*: number of experiments.

## 3. Results

### 3.1. Comparison of 2D and 3D DRG Explant Culture Models

To compare 2D and 3D models, mouse DRGs were isolated and cultured on a Matrigel surface or in a Matrigel drop, respectively. On the third day, the images of the DRGs with the outgrowing neurites were captured using a light microscope with a 5× objective. In 2D explant cultures, no neurites were formed, whereas in 3D cultures, extensive neurite outgrowth was detected (Figure 4). Immunofluorescent staining of 2D DRG explants are difficult to perform because DRGs adhere poorly to Matrigel and can be easily lost during the staining and washing procedures. Length assessment of the outgrowing axons and measurement of glial cell migration without attached DRGs can lead to misinterpretation of the results. Our obtained results are in accordance with the previously published data because only glial cells migrated from DRGs in 2D cultures. However, no outgrowing neurites could be detected [23].

Thus, the 3D explant culture represents a relevant model resembling in vivo regeneration, i.e., the outgrowth of neurites, which can be subjected to qualitative and quantitative analysis.

### 3.2. GDNF Stimulated Axon Regeneration in 3D Model of DRG

DRG isolation involves axon damage and ganglion membrane rupture, followed by axon regrowth in culture conditions. For standardization of the protocol and correct comparison of axon outgrowth, the DRGs should be obtained from the same parts of the spine, as DRGs from different vertebra segments differ in size and neuronal composition. In addition, for animals of the same age and gender, the same Matrigel volumes are recommended for a study. Evaluation and comparison of the neurite outgrowth and cell migration between experimental groups should be carried out the same day.

Mouse DRGs were isolated from the thoracic spine and placed in a drop of Matrigel. After Matrigel polymerization, the culture media containing NGF or GDNF or BDNF or 10% BSA as a control was added. On the 14th day, DRGs were immunofluorescently stained with antibody against NF-200 (a marker of large axons [24]). Nestin-GFP-positive cells (green fluorescence) and NF-200 antibody staining (red fluorescence) were visualized using confocal microscopy. GDNF exerted the most prominent effect on axon regrowth (Figure 5A)—upon addition of GDNF, the average number of axons in a field of view was twofold greater than in control (BSA) (*p* < 0.05, *n* = 4). No statistically significant difference in the number of axons between BDNF or NGF groups was detected compared to the control (Figure 5B).

### 3.3. GDNF and NGF Stimulated Nestin-GFP Positive Cell Migration

It is well known that activated glial cells are indispensable for axon regeneration in the PNS [25]. In the current study, using Nestin-GFP-expressing mice in which Nestin-positive neural precursors expressed GFP, we evaluated the number of GFP-positive cells that migrated from the DRG into Matrigel (Figure 6A). We found that NGF and GDNF, but not BDNF, increased Nestin-GFP cell migration threefold and twofold, correspondingly, compared to the control (*p* < 0.05) (Figure 6B).

### 3.4. Nestin-Expressing Cells Expressed GFAP Glial Fibrillary Acidic Protein

To test the differentiation potential of Nestin-expressing cells, we obtained primary cell cultures from DRGs isolated from Nestin-GFP mice and carried out immunofluorescent staining with antibody against GFAP (a marker of immature Schwann cells (glial cells)). Neurons were stained using antibody against NF200. Confocal microscopy analysis revealed that all Nestin-expressing cells also expressed GFAP (Figure 7), whereas NF200-positive neurons did not express Nestin (Figure 8). These data indicate that Nestin-expressing cells are glial cell precursors.

## 4. Discussion

Numerous studies indicate the important role of neurotrophic factors in neuronal survival and axon growth in the developing and adult nerve systems [26]. Following nerve injury, expression and activity of certain neurotrophic factors (NGF, BDNF, and GDNF) increases, stimulating the growth of regenerating axons and the reinnervation of the target organs [27]. After binding to the specific receptors on the cell surface of neurons, neurotrophic factors are internalized and transported to the cell body by retrograde axonal transport, resulting in activation of signaling cascades that trigger neuron survival [28].

The first identified and most thoroughly studied neurotrophin is NGF [29]. NGF and its receptors (p75NTR and TrkA) are abundantly expressed during embryogenesis long before the onset of neural tube formation and neuronal differentiation [30]. NGF plays an essential role in the survival and differentiation of sympathetic neurons and subpopulations of sensory neurons in embryogenesis, as well as in the regulation of gene expression associated with axonal growth and synaptogenesis [29]. In adults NGF is involved in phenotype maintenance of sympathetic and sensory neurons in the PNS and for the functional integrity of forebrain cholinergic neurons in the central nervous system (CNS) [29,30,31]. It was shown that NGF expression decreases at 6 h after peripheral nerve injury due to the blockage of NGF retrograde transport, however, it recovers by the second day due to the increased synthesis of NGF in DRGs [32,33]. In rats, transection of the sciatic nerve is followed by a substantial increase in NGF content as a result of the up-regulated NGF synthesis in non-neuronal cells, which is important for axon regeneration [32]. In our 3D explant model, NGF stimulated Nestin-GFP cell migration from DRG into the Matrigel, but had no effect on the outgrowth of axons. 

Nestin is related to a family of intermediate filaments and is expressed in the neural stem cells and precursor cells that give rise to a wide spectrum of cell types in the developing and adult nervous systems [34]. Earlier it was reported that Nestin-expressing cells are important for functional nerve repair [35]. Being transplanted into the injured sciatic nerve or spinal cord, Nestin-expressing cells differentiate into Schwann cells, which are known to support axon regeneration [35]. Following peripheral nerve injury, Schwann cells revert to an immature state capable of proliferating, migrating, and forming the so-called “Büngner bands”, thus ensuring directed growth of regenerating axons towards denervated targets [19]. Schwann cells also synthesize and secrete neurotrophic factors, in particular, NGF and BDNF, which increase the survival of neurons and stimulate the growth of axons [36].

In embryogenesis, BDNF is expressed in a number of cells in the hippocampus, in the piriform cortex, the thalamus, the hypothalamus, and the amygdale by certain cells in the cortex, and is not expressed in the striatum [37]. In adults, BDNF is expressed ubiquitously throughout the brain [38]. In the central and peripheral nervous systems, BDNF ensures the survival of several types of neurons, such as mesencephalic dopaminergic neurons, septal cholinergic neurons, striatal GABAergic neurons, retinal ganglion neurons, cerebellar neurons, and motor neurons. BDNF also promotes differentiation of the central cholinergic and dopaminergic neurons [39]. In 2001, Boyd and Gordon reported no significant effects of BDNF on axon regeneration [40]; however, later in 2003, combined administration of BDNF and GDNF synergistically enhanced motor axon regeneration in a chronic trauma model [41]. In the present work, we detected no effect of BDNF administration on axon regeneration or Nestin-GFP cell migration. One of the possible explanations is that BDNF, being ubiquitously expressed in the adult brain with the highest level in the hippocampus, cerebellum, cerebral cortex, hypothalamus, and septum [42], exerts its effects on axon regeneration mainly in the CNS, rather than in the PNS.

GDNF promotes the survival of various populations of neurons at different stages of their development in the PNS and CNS [43]. GDNF is rapidly upregulated in denervated Schwann cells after sciatic nerve injury, and is retrogradely transported by motoneurons to the cell body [41]. GDNF expression in the denervated distal nerve stump and the expression of GDNF-family receptor α-1 (GFRα-1) in axotomized motoneurons are consistent with a functional role for GDNF in motor axonal regeneration [41]. Sustained GDNF delivery to the site of nerve injury appeared to be more effective than NGF in regeneration of both sensory and motor axons over long gaps [44]. In the present work, GDNF administration into the culture media resulted in a twofold increase in the number of axons, and enhanced the migration of Nestin-GFP-expressing cells from the DRG explants into the Matrigel. It is known that high-dose administration of GDNF in adult rats stimulates Schwann cell proliferation and axon myelination, even those that were initially non-myelinated [44]. These data suggest that GDNF not only has a trophic function for axons and Schwann cells, but can also mediate axon–glial interactions.

It should be mentioned that the 3D DRG model described in the present study has practical applications and allows the dynamic evaluation of various parameters for 21 days in culture. Using this model, we assessed the number and the length of the forming neurites, the rate and direction of neurite growth, the increased/decreased neurite branching, the migration and chemotaxis of neural cells, and the attractive or repulsive effects of the agents introduced into the Matrigel. In addition, the described method allowed us to carry out live observation of neurite outgrowth and to visualize the glial cell migration from the DRGs in real time. Moreover, the isolation of DRGs from Nestin-GFP mice ensured specific visualization of glial cells, as DRGs may contain non-neuronal cells (fibroblasts, etc.) that can be also identified using immunofluorescent staining procedures.

The 3D DRG model can be applied to screen potential therapeutic drugs for the aims of regenerative medicine. The immunofluorescence staining technique for axon visualization can be used to trace the pattern of axon growth and branching. Therefore, the developed method is applicable in fundamental studies for the search for mechanisms to stimulate nerve regeneration, as well as in R&D studies for the development of new drugs.

In the present study, the method was used to test the impact of growth factors (NGF, BDNF, GDNF) on axon outgrowth and neural cell migration from the DRGs. On the basis of the aforementioned finding, we conclude that GDNF had the greatest stimulating impetus on both axon formation and glial cell migration. Although NGF significantly affected glial cell migration, its effect on axon growth was insufficient for axon outgrowth.

## Figures and Tables

**Figure 1 biomedicines-08-00049-f001:**
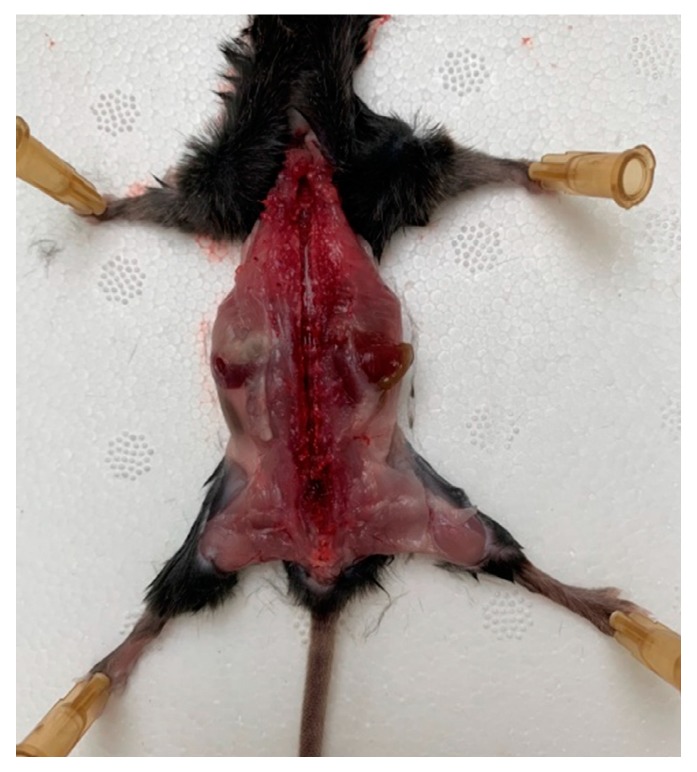
A mouse on the operating table with the limbs maximally distant from the body, the spinal column released from the muscle, and connective tissues and spinal canal opened.

**Figure 2 biomedicines-08-00049-f002:**
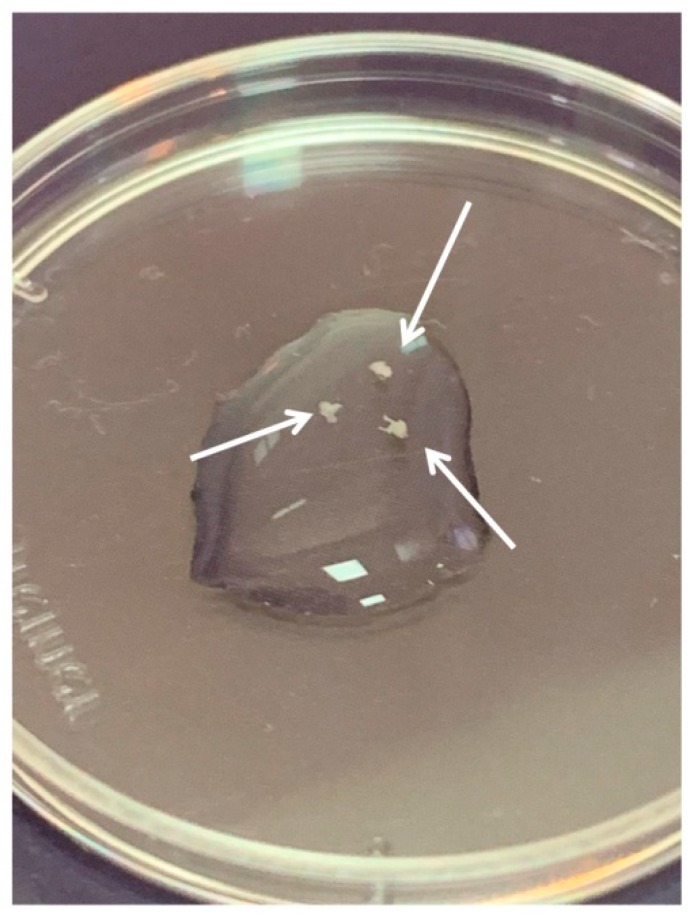
Three dorsal root ganglia (DRGs) in HBSS. Arrows point to the DRGs.

**Figure 3 biomedicines-08-00049-f003:**
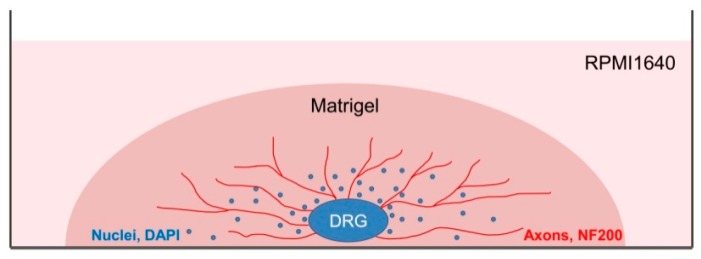
A scheme presenting DRG explant cultured in 3D conditions. Each DRG was placed in a well of an 8-well plate and a drop of Matrigel (60 µL) was used to cover each DRG. After Matrigel polymerization, RPMI culture media was added to each well. On the 14th day, DRGs were immunofluoresently stained using antibody against NF200 for axon visualization and counterstained with DAPI to visualize the nuclei.

**Figure 4 biomedicines-08-00049-f004:**
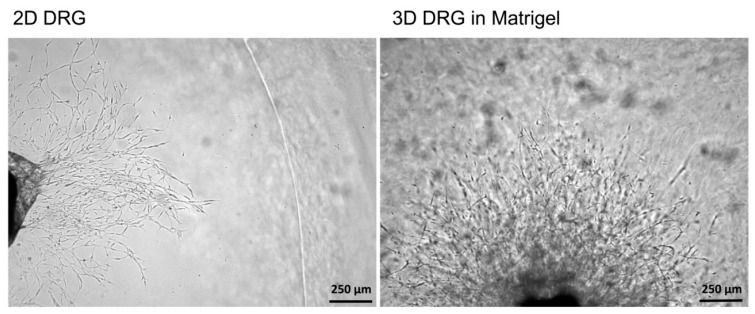
A comparison of 2D and 3D models of DRG in Matrigel ex vivo. In 2D culturing conditions, cells migrating from the DRG were noted, however, no neurites growing from DRGs were detected. In 3D cultures, cell migration was accompanied by the outgrowing neurites.

**Figure 5 biomedicines-08-00049-f005:**
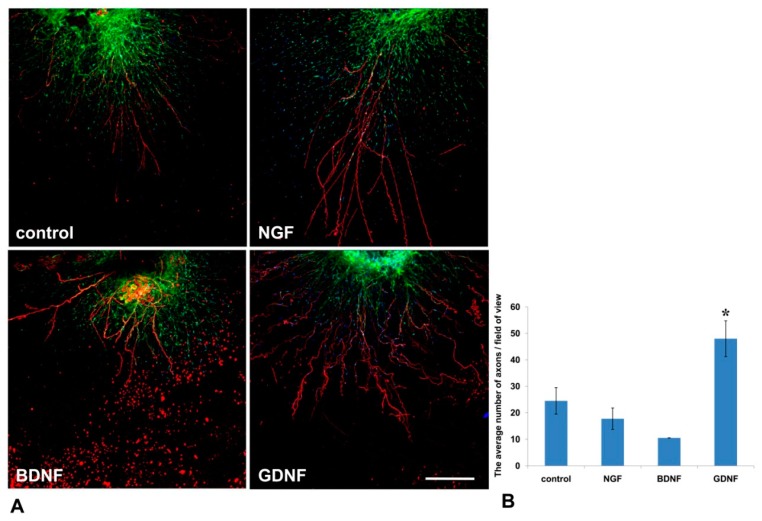
GDNF stimulated axon outgrowth from DRGs into Matrigel in the 3D explant model. NFG, GDNF, or BDNF was added to the culture media. (**A**) Immunofluorescent staining of mouse DRG in a Matrigel drop. Axons were visualized with antibody staining against NF200 (red fluorescence), nuclei counterstained with DAPI, and Nestin-positive cells (green fluorescence). Scale bar: 250 µm. (**B**) A graph presenting the quantification of axon growth. Data are presented as mean ± standard deviation (SD). * *p* < 0.05, *n* = 4.

**Figure 6 biomedicines-08-00049-f006:**
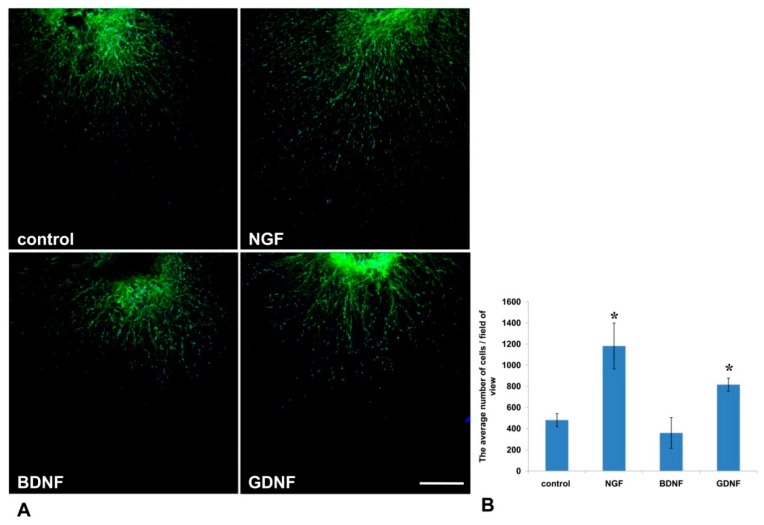
GDNF and NGF stimulated migration of Nestin-green fluorescent protein (GFP) cells from DRGs into the Matrigel in a 3D explant model. (**A**) A microphotograph presents the DRG from Nestin-GFP mouse in a drop of Matrigel. Nuclei are counterstained with DAPI, and green fluorescence corresponds with Nestin-GFP-expressing cells. Scale bar: 250 µm. (**B**) A graph presenting the average number of double-positive (GFP and DAPI) cells that migrated into the Matrigel. Data are presented mean ± standard deviation (SD), * *p* < 0.05, *n* = 4.

**Figure 7 biomedicines-08-00049-f007:**
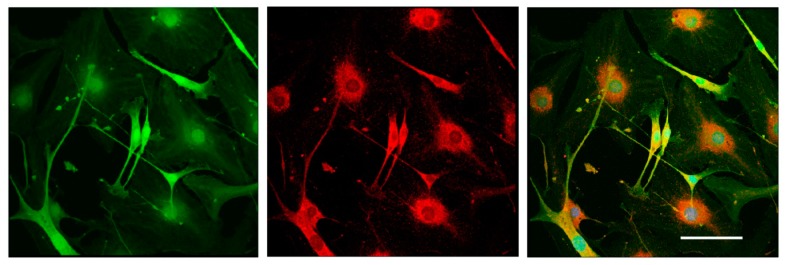
Immunofluorescent staining of glial cells derived from mouse DRGs after 3 days in vitro. Cells were stained with antibody against GFAP (red fluorescence). Nuclei were counterstained with DAPI, and green fluorescence corresponds to Nestin-GFP-expressing cells. Scale bar: 150 µm. Yellow color corresponds to merged green and red fluorescence.

**Figure 8 biomedicines-08-00049-f008:**
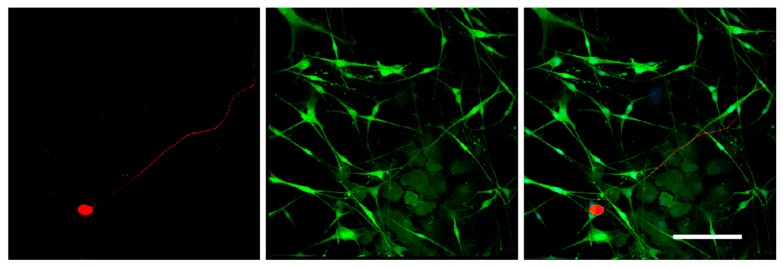
Immunofluorescent staining of the primary cell culture derived from mouse DRGs after 3 days in vitro. Cells were stained with antibodies against NF200 (red fluorescence). Nuclei were counterstained with DAPI, and green fluorescence corresponds to Nestin-GFP-expressing cells. Scale bar: 200 µm.

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
