# Peer review of "Three-Dimensional Model of Dorsal Root Ganglion Explant as a Method of Studying Neurotrophic Factors in Regenerative Medicine"

_biomedicines, 2020, doi:10.3390/biomedicines8030049_

Round 1

Reviewer 1 Report

Klimovich et al. have performed additional experiments in response to some of my concerns. The manuscript has been strengthened by these additions and I can now recommend the revised version for publication in Biomedicines.

I have only one remaining question however, which would ideally be clarified prior to publication. Could the authors confirm that the unit for a drop of Matrigel is um (Figure 3)?

Author Response

Point 1: I have only one remaining question however, which would ideally be clarified prior to publication. Could the authors confirm that the unit for a drop of Matrigel is um (Figure 3)?

Response 1: This is a mistake. The drop of Matrigel is 60 ul. We will correct it in the manuscript. 

Reviewer 2 Report

The differences between 2d and 3d cultures show differences not only in the neurological field but also in others (see reference).
The paragraph with reagents and antibodies should be separated and the individual products inserted within each paragraph according to the test carried out

Author Response

Point 1: The differences between 2d and 3d cultures show differences not only in the neurological field but also in others (see reference).

Response 1: We would like to thank the Reviewer for the comment. Indeed, differences in 2D and 3D cultivation are applicable to different types of cells and tissues, which we describe in the introduction. This method can be used not only in the neurological field but also for cardiology, cancer biology etc.

Point 2: The paragraph with reagents and antibodies should be separated and the individual products inserted within each paragraph according to the test carried out

Response 2: The paragraph with reagents and antibodies was deleted. Reagents and antibodies were inserted within each paragraph according to the procedure.

Round 2

Reviewer 2 Report

add bibliographic references related to other fields of applications (tissues, cells). Refer to the bibliographic note inserted in the comments during the last revision but also other notes

Author Response

Dear Sir/Madam, We agree with your comment and would be very glad to add the suggested references into the manuscript, however, neither we nor the editorial board can find any notes in the revised version of the manuscript. We are kindly asking you to send them directly  into the reviewing form.

This manuscript is a resubmission of an earlier submission. The following is a list of the peer review reports and author responses from that submission.

Round 1

Reviewer 1 Report

In this manuscript, Klimovich et al. investigate the effects of neurotrophic factors on axon growth and glial cell migration using dorsal root ganglia (DRG) explants. DRG explant is a three-dimensional (3D) model system, which maintains the complexity of neural tissues and is widely used to study axon regeneration in the field. The authors illustrate the experimental procedures of culturing DRG explants, and further apply this model to assess axon regeneration ability of three neurotrophic factors.

DRG explant model using different embedding material (e.g. Matrigel, hydrogel, and agarose gel) has been previously reported, and promoting axon outgrowth by adding neurotrophic factors in the culture media is also well-characterized, which make this manuscript less novel. Unfortunately, I regret that I cannot recommend this manuscript for publication in Biomedicines, and I have a number of comments/suggestions outlined below that should help to clarify the findings and improve the manuscript.

The bright-field images shown in Figure 4 are very difficult to judge the axon outgrowth/glial migration between 2D and 3D culture. Performing immunolabeling of axonal marker or showing the GFP signal would greatly strengthen the statement.

In Figure 6, the authors demonstrate that GDNF and NGF increase the migration of GFP-positive cells (neural stem cells or neural precursor cells) from DRG explants into Matrigel. Given the important role for neurotrophic factors in cell proliferation, it remains unclear whether the increase in GFP-positive cells is due to increased cell migration and/or cell proliferation.

The arrows shown in Figure 2 do not point to the DRGs, and the unit for Matrigel’s volume should be 60 uL instead of uL. Moreover, it would be more suitable to present GFP-nestin cells as Nestin-GFP cells, which the expression of GFP transgene is controlled by the regulatory elements of the Nestin gene.

In the introduction, the authors mention that “Primary neural cell cultures represent a modern challenge in cell biology, since mature neurons do not undergo cell division in vitro, which makes it complicated to obtain a sufficient number of cells for experiments [13]. Cell survival is usually low and such cultures can be maintained for only 5-7 days, provided that soluble factors that increase survival are added [14].” It is known that primary cortical neurons could be maintained at least for 2 weeks in vitro. Do the authors refer specifically to DRG explants or to general primary neuronal culture?

Reviewer 2 Report

In the present article, the authors developed a 3D murine dorsal root ganglia (DRG) model using Matrigel. The authors found that GDNF induced the most extended axon growth, compared to the other neurotrophic factors (NGF and BDNF) using the developed 3D DRG model. Additionally, the authors found that NGF and GDNF stimulated the migration of the GFP-nestin positive cells more, compared to BDNF, with their 3D DRG model. Those findings are valuable; however, the simple comparison of the effectiveness of neurotrophic factors on the growth of DRG can not suggest any possible strategy to improve the neural tissue regeneration following nerve injury. More importantly, the rationale, why the comparative study among the different types of neurotrophic factors is required, is missing in the present article. Besides, the 3D culture of neural tissue with Matrigel is not novel; it needs to clarify which part of the developed model is the improvement compared to other current 3D culture approaches of the peripheral nerve system. Moreover, the authors must address why the developed model can represent peripheral nerve regeneration after injury. For those reasons, I suggest 'Reject' for the present article.